# Host Cell Rap1b mediates cAMP-dependent invasion by *Trypanosoma cruzi*

**Gabriel Ferri**[1,2], **Daniel Musikant**[1,2], **Martin M. Edreira**[1,2]*

**1** CONICET-Universidad de Buenos Aires, IQUIBICEN, Ciudad de Buenos Aires, Argentina, **2** Laboratorio de Biología Molecular de Trypanosomas, Departamento de Química Biológica, Facultad de Ciencias Exactas y Naturales, Universidad de Buenos, Ciudad de Buenos Aires, Argentina

* mme2@pitt.edu

## Abstract

*Trypanosoma cruzi* cAMP-mediated invasion has long been described, however, the detailed mechanism of action of the pathway activated by this cyclic nucleotide still remains unknown. We have recently demonstrated a crucial role for Epac in the cAMP-mediated invasion of the host cell. In this work, we gathered evidence indicating that the cAMP/Epac pathway is activated in different cells lines. In accordance, data collected from pull-down experiments designed to identify only the active form of Rap1b (Rap1b-GTP), and infection assays using cells transfected with a constitutively active mutant of Rap1b (Rap1b-G12V), strongly suggest the participation of Rap1b as mediator of the pathway. In addition to the activation of this small GTPase, fluorescence microscopy allowed us to demonstrate the relocalization of Rap1b to the entry site of the parasite. Moreover, phospho-mimetic and non-phosphorylable mutants of Rap1b were used to demonstrate a PKA-dependent antagonistic effect on the pathway, by phosphorylation of Rap1b, and potentially of Epac. Finally, Western Blot analysis was used to determine the involvement of the MEK/ERK signalling downstream of cAMP/Epac/Rap1b-mediated invasion.

## Author summary

As an obligate intracellular parasite, *T. cruzi* needs to gain access to the host cell cytosol, where replication and differentiation takes place. The process of invasion is mediated by the activation of different signaling pathways in the host. Activation of cAMP signaling has been implicated in invasion, with high intracellular levels of cAMP associated with an increase in invasion. In this regard, we have recently shown that Epac (Exchange protein activated by cAMP), and not PKA, mediates the effect of the cyclic nucleotide on parasite internalization. In this work, in order to shed some light into the detailed mechanism of invasion mediated by cAMP/Epac, we have explored the activation of downstream effectors of this pathway during host cell infection. Our results established that Rap1b, a known effector of Epac, and MEK/ERK, are downstream effectors of the pathway. In addition, we found a PKA-dependent inhibition of infection, presumably by phosphorylation of Rap1b, and potentially of Epac.

**Data Availability Statement:** All relevant data are within the manuscript and its Supporting Information files.

**Funding:** This work was partially supported by the Agencia Nacional de Promoción Científica y

Tecnológica (ANPCyT, Argentina) grant PICT-2015-1713 to MME. The funders had no role in study design, data collection and analysis, decision to publish, or preparation of the manuscript.

**Competing interests:** The authors have declared that no competing interests exist.

## Introduction

As an obligate intracellular parasite, *Trypanosoma cruzi* replicates in the cytoplasm of infected mammalian host cells. The ability to infect host cells depends on the stage of the parasite, the strain and the DTU (Discrete Typing Units), as the result of the expression of a differential repertoire of surface/secreted molecules, that activates different signalling pathways in the host cell. In general, the attachment of trypomastigotes activates several hosts signalling pathways, including the elevation of intracellular cAMP levels in the host cell [1]. It has been shown that invasion involves the recruitment and fusion of lysosomes to the entry site [2], and that cAMP potentiates the $Ca^{2+}$-dependent exocytosis of lysosomes and lysosome-mediated cell invasion [3]. Although a transient increase of $Ca^{2+}$ and the recruitment of lysosomes are common features in the invasion of metacyclic trypomastigotes (MTs) and tissue culture-derived trypomastigotes (TCTs) [4,5], the signalling pathways that these parasites promote in the host cell are different. Among them, the activation of cAMP-mediated signalling by TCTs is a poorly studied process. It has been previously demonstrated that the pharmacologic intervention of the cAMP pathway was able to modulate parasite invasion [4,6,7]. To determine the specific role of cAMP main effectors, PKA and Epac, in *T. cruzi* invasion, we used a set of pharmacological tools to selectively activate or inhibit these proteins. Whereas differential activation of PKA had no effect on invasion, a significant increase in invasion was observed in cells treated with a cAMP analogue that selectively and exclusively activates Epac [7]. Accordingly, the inhibition of Epac by ESI-09 [8] showed a significant decrease in invasion. Unexpectedly, specific inhibition of PKA also showed a positive effect on invasion, suggesting a PKA/Epac crosstalk during the process of invasion [7]. In this regard, it has been described that both proteins can be recruited to the same microdomain through the association with radixin [9,10], an ERM structural protein that attaches the plasma membrane to the cortical actin cytoskeleton [11]. Respectively, confocal studies have shown that ERM proteins are associated with the invasion site of extracellular amastigotes (EAs), where colocalize with F-actin [12]. Moreover, a link between radixin and the cAMP/Epac-dependent pathway during TCT invasion was confirmed by blocking host cell invasion with a permeable version of 15-mer sequence (stearate-KPRAC-SYDLLLEHQRP-amide peptide) corresponding to the minimal Epac1 ERM binding domain. This peptide displaces the Epac protein from its association with radixin and delocalized it from the microdomain. Under these conditions, the percentage of invasion is similar to that obtained when the Epac protein is inhibited by ESI-09 [7]. Taken together, these results clearly established a crucial role for Epac in the cAMP-mediated invasion of the host cell.

However, downstream effectors involved in this pathway are still unknown. Epac1 has been involved in PI3K/Akt and MEK/ERK pathways [13,14], and members of these pathways, including Rap1, were localized at late endosomes/lysosomes [15]. In cardiomyocytes, the cAMP/Epac/Rap1 pathway modulates the excitation-contraction mechanism by stimulating $Ca^{2+}$ release through ryanodine receptors (RyR) [16]. In smooth muscle cells, Rap1 inhibits RhoA activity and promotes $Ca^{2+}$ desensitization and smooth muscle relaxation [17]. Rap1 activation also induces muscle hyperpolarization by decreasing $Ca^{2+}$ influx by inducing the opening of $Ca^{2+}$-sensitive $K^+$ channels, generating a boost in vasodilation [18]. In addition, Rap1 was shown to modulate mitogen-activated kinases (MAPKs), like extracellular signal-regulated kinase (ERK1/2), inducing the stimulation or inhibition of these kinases depending on the cell type. More recently, the role of Rap1 in ERK phosphorylation and activation in smooth muscle was demonstrated [19]. Ral-GDS, an effector of Rap1, promotes cardiomyocyte autophagy [20], and the downstream effector of Ral-GDS, RalB, binds specific subunits of the exocytosis machinery and mediates activation of autophagosome assembly [21].

Smooth muscle and heart are the most important target organs for *T. cruzi* infection and persistence during the chronic phase of Chagas disease. Taking into account that Epac has a critical role in cAMP-mediated invasion and the regulation of various cAMP-dependent functions in smooth muscle and heart, possibly modulating the intracellular concentration of $Ca^{2+}$ through the activation of Rap1 and the participation of ERK1/2 [18,22,23], deciphering the detailed functioning of the cAMP/Epac pathway would provide a deeper insight into the host cell invasion mechanisms mediated by this cyclic nucleotide. In this work, we investigated the involvement of two known effectors, Rap1b and ERK, as potential mediators in the cAMP/Epac-dependent invasion by *T. cruzi* and the role of PKA-dependent Rap1b phosphorylation.

## Materials and methods

### Cells and parasites

NRK (ATCC CRL-6509), VERO (ATCC CCL-81) and HELA (ATCC CCL-2) cell lines were cultured in DMEM medium supplemented with Glutamax (Gibco), 10% (v/v) FBS (Natocor), 100 U/ml penicillin and 0.1 mg/ml streptomycin (Sigma), and maintained at 37˚C in a 5% $CO_2$ atmosphere. The HL-1 cell line [24] was cultured in a gelatin/fibronectin matrix (5 μg fibronectin / 0.02% gelatin (m/v)-Sigma) and Claycomb culture medium supplemented with Glutamax(Gibco), 10% (v/v) FBS, 100 U/ml penicillin, 0.1 mg/ml streptomycin and 0.1 mM norepinephrine (Sigma). Tissue culture-derived trypomastigotes forms (TCT) of *T. cruzi* Y strain were routinely maintained in VERO cells cultured in DMEM supplemented with 4% FBS and penicillin/streptomycin. Trypomastigotes were obtained from supernatants of infected VERO cells by centrifugation. First, the supernatant conditioned medium was centrifuged at low speed (500 g) to remove intact cells and cell debris Then, the supernatant obtained was centrifugated at 3,000 g for 15 min. and the pellet with the parasites was washed in PBS three times.

### Invasion assay

Cells were grown on glass cover slides in a 24 multi-well plate with DMEM 10% FBS for 24 hours at $2x10^4$ cells/well density at 37˚C, 5% $CO_2$ and incubated with: 37,5 μM of the Epac1 inhibitor ESI-09 (Sigma); 300 μM of 8-Br-cAMP (Biolog); 50 μM of the MEK1/2 inhbitor PD98059; or 0,1% DMSO as a control condition. Cells were then washed and infected with trypomastigotes of the Y strain (moi 100:1) for 2 hours. Parasite were removed and cells incubated for 48 hs. Cells were fixed, stained with DAPI and infection level determined by fluorescence microscopy. Percentage of infection (calculated as the (#Infected cells/total counted cells)*100) and amastigotes/100 cells were calculated counting 3,000 cells, expressed as mean ± SD of three or more independent experiments and performed in triplicate. Infection of non-treated cells was considered as basal infection.

### Host cell transfection

A transient transfection protocol with polyethyleneimine (PEI) was used [25]. Briefly, cells were grown at about 60% confluence and incubated at 37˚C in a 5% $CO_2$, 95% humidified air environment. Next day, cells were transfected with pCGN empty vector (EMPTY), pCGN-HA-Rap1b (HA-Rap1) or HA-Rap1b mutants (G12V, S179A, S179D, or combinations) (kindly provided by Dr D. Altschuler, University of Pittsburgh, USA) using a ratio of 4:1 PEI: DNA mix in OptiMEM medium (Gibco). The mixture was kept for 30 min. at room temperature and then added to the cells and incubated at 37 C and 5% $CO_2$. After 24h, cells were

washed with PBS and complete medium (DMEM or Claycomb 10% FBS) was added. The transfected cells were used at 24h post-transfection.

## Trypomastigote release assay

HL-1 cells were seeded on a 24-well plate at a concentration of 7000 cells/mL in Claycomb medium supplemented with 10% FBS. After 24 hours, cells were infected and treated as described above. 72 hours later, medium was replaced with fresh prepared treatments until trypomastigotes were observed under microscope at six days post infection (pi). Supernatants were transferred to a new plate, washed to avoid mammalian cell contamination and a solution of resazurin sodium salt was added as a fluorogenic oxidation-reduction indicator (final concentration 0.1 mM). After 3 hours of incubation, fluorescence was measured with a FLUOstar OPTIMA (BMG LABTECH) microplate reader at 590 nm (excitation: 570 nm). Baseline corrected values of fluorescence were normalized to the negative control. Results are expressed as mean ± SD of three or more independent experiments and performed in triplicate.

## GST Pull-down

Detection of active Rap1 (GTP-bound) was performed through pull-down assays using a recombinant GST-RBD protein (GST fusion to the Rap1b-binding domain of the RalGDS protein, which only recognizes active Rap). A total of 1 mL bacteria lysates containing GST or GST-RBD were mixed by rotation with 40 μl 50% GSH-Sepharose at 4°C for 1 h. The beads were centrifuged at 800 g for 2 min. at 4°C and washed with lysis buffer. Lysates from HA-Rap1 transfected cells pre-treated for 2h with 8Br-cAMP, infected with trypomastigotes of the Y strain (Tp Y) or mock infected (Ctrl) were incubated with RBD-glutathione-agarose resin for 1h at 4°C. Resin was washed and eluted with cracking buffer for WB analysis.

## Western Blot

After electrophoresis, the gel was equilibrated in 25 mM Trizma base, 192 mM L-1 glycine and 20% v/v methanol pH 8.3. Then, proteins were transferred to previously hydrated with methanol PVDF membranes (Amersham Hybond, GE Healthcare) in a vertical tank (Mini-PROTEAN Tetra Cell, Bio-Rad). After transfer, membranes were blocked with 20 mM L−1 Tris-HCl, 500 mM NaCl, 0.05% Tween and 5% non-fat milk, pH 7.5, incubated with anti-GST (Genscript), anti-p44/42 MAPK (ERK1/2, Cell Signaling), anti-phospho-p44/42 MAPK (pERK1/2, Cell Signaling) or anti-GAPDH (Santa Cruz Biotechnology) antibodies. After incubation, membrane was washed and incubated with rabbit horseradish peroxidase (HRP)-IgGs antibody (Santa Cruz Biotechnology), washed again and then revealed using 0.88 mg/ml luminol, 0.066 mg/ml p-coumaric acid, 6 mM $H_2O_2$; 100 mM Tris-HCl, pH 8.8 solution. Chemiluminescence was recorded with the C-DiGit scanner (LI-COR), and bands intensity were quantified with ImageJ and ImageLab 6.1 (Bio-Rad) software.

## ERK phosphorylation

Cells were treated with or without PD98059 for 2h and incubated with trypomastigotes of the Y strain for 30 min., treated with 750 μM $H_2O_2$ for 5 min. or mock infected. Then, cells were lysed and cracking buffer added for WB analysis.

## Indirect immunofluorescence assay

Cells were adhered to glass previously treated with 40 μg/ml of poly-D-lysine (Sigma), fixed with PBS-PFA 4% (Sigma), washed with PBS and incubated with $NH_4Cl$ for 15 min. Then,

were permeabilized with 0.2% Triton-x100 and incubated with anti-RAP1 antibody (Genscript) at 4°C. After 16 h, washed with PBS, incubated with mouse anti-IgG (H+L) anti-conjugated to Alexa Fluor594 antibody (Jackson InmunoResearch), and nuclei stained with DAPI. Finally, glasses were mounted on slides with FluorSave mounting solution (Merk Millipore). Preparations were analysed in a Nikon Eclipse E600 fluorescence microscope.

## Results

### cAMP/Epac activation as a ubiquitous mechanism of invasion in *T. cruzi*

The crucial role of Epac during invasion by *T. cruzi* was recently described in NRK cells [7]. In order to assess the ubiquity of the cAMP/Epac pathway, other cell lines were used in invasion assays. Similar to what happened in NRK cells [4,7], high levels of cAMP induced by a non-hydrolysable permeable analogue of cAMP, 8-Br-cAMP (8-Bromoadenosine 3′,5′-cyclic monophosphate) (Biolog), positively modulated invasion in both HELA and HL-1 cells (Fig 1). Consistent with this result, specific pharmacological inhibition of Epac by ESI-09 (Sigma), resulted in a significant decrease in invasion in both cell lines (Fig 1).

### Rap1b as a mediator of the cAMP/Epac-dependent invasion

The participation of Rap1b as mediator of the cAMP/Epac-dependent invasion was evaluated by pull-down experiments using agarose bound GST-RalGDS Rap-binding domain (GST-RBD), designed to pull down only the active form of Rap1b (Rap1b-GTP). Briefly, cells transfected with HA-Rap1b were incubated in the presence of DMSO (vehicle), 8-Br-cAMP, ESI-09 or trypomastigotes of Y strain of *T. cruzi* for 2 h. Cells were lysed and lysates used in pull-down experiments (Figs 2 and S1 and S2). As shown in Fig 2, higher levels of activated GTP-bound Rap1 were detected in lysates from cells incubated with 8-Br-cAMP and trypomastigotes, while inhibition of Epac by ESI-09 reduced Rap1b activation as expected (S2 Fig), supporting the involvement of Rap1b in cAMP-mediated invasion.

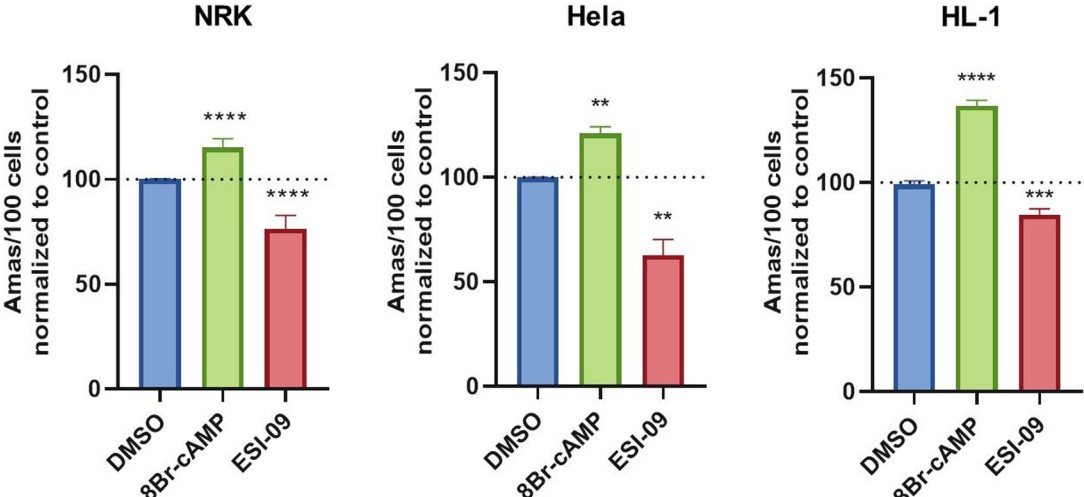

**Fig 1. cAMP/Epac pathway is required for *T. cruzi* invasion in different cell lines.** Pre-treated NRK, HELA or HL-1 cells (30 min at 300 µM of 8-Br-cAMP or 37.5 µM of ESI-09) were infected with trypomastigotes from *T. cruzi* Y strain (100:1 parasite to cell ratio for 2 h). 48 hs post-infection cells were fixed, stained with DAPI and percentage of infection determined by fluorescence microscopy. Infection of untreated cells was considered as basal infection. Results are expressed as mean ± SD (n ≥ 3). **** p<0.0001, *** p <0.001, ** p <0.01; ANOVA and Dunnett's post-test.

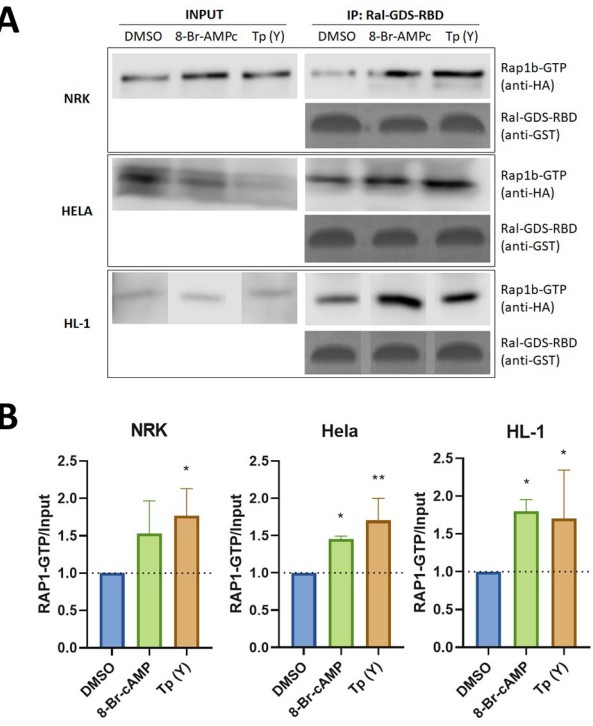

**Fig 2. Rap1b pull-down assays. A)** HA-Rap1 transfected NRK, HELA or HL-1 cells were incubated for 2 h with 8-Br-cAMP, infected with trypomastigotes from *T. cruzi* Y strain or mock infected (DMSO). Then, cells were lysed and pull-down assay with glutathione-agarose resin performed for 1h at 4°C. Resin was washed and eluted with cracking buffer for WB analysis. **B)** Bands were quantified and normalized against the input using ImageJ cell software. Results are expressed as mean ± SD (n≥3). * p<0.05, ** p <0.005, One-way ANOVA–Dunnett´s multiple comparison test.

In accordance with these results, when HELA and HL-1 cells transfected with a constitutively active mutant of Rap1b, Rap1b-G12V, in which a single point substitution, glycine-to-valine at codon 12 of Rap1b, a significant increase in infection was observed when compared with the control (Fig 3A–3D, respectively). Noteworthy, when the complete invasion-differentiation-release cycle was evaluated in HL-1 cells overexpressing Rap1b-G12V (Fig 3E), trypomastigotes released into the medium showed similar results than the results obtained for percentage of infected cells and amastigotes/100cells (Fig 3C and 3D), suggesting the cAMP/Epac/Rap1b pathway would play a role in the early steps of the establishment of the infection, as previously hypothesized [7]. In addition to these results suggesting that Rap1b-GTP is required as a mediator of the cAMP/Epac1 pathway during the invasion by *T. cruzi*, fluorescence microscopy in HL-1 (Figs 4 and S3) and Hela cells (S4 Fig), revealed the relocalization of Rap1b, reflected as an increase in the fluorescence intensity of Rap1b, to the site of entry of *T. cruzi*, supporting the hypothesis that Rap1b needs to be activated and properly localized in the entry site.

## Role of PKA-dependent Rap1b phosphorylation

While the specific activation of PKA had no effect on invasion, an increase in internalized parasites was observed as a result of PKA inhibition [7]. Therefore, under physiological conditions, PKA-mediated phosphorylation would negatively regulate the cAMP/Epac pathway of invasion. The inhibition of the Epac-mediated invasion pathway could be achieved, at least, at two different levels: through direct phosphorylation of Epac or at the level of Rap1, an Epac

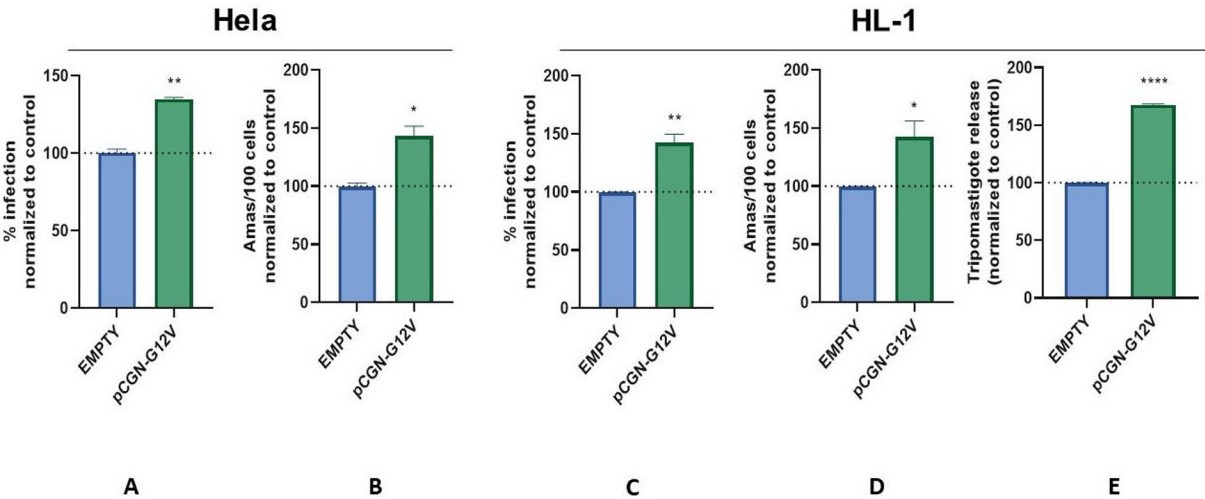

**Fig 3. Invasion assays.** Rap1 transfected Hela and HL-1 cells were infected with trypomastigotes from *T. cruzi* Y strain (100:1 parasite to cell ratio for 2 h). 48 hs post-infection cells were fixed, stained with DAPI and percentage of invasion determined by fluorescence microscopy. Infection of untreated cells was considered as basal infection. Percentage of infection (calculated as the (#Infected cells/total counted cells)*100) and amastigotes/100 cells were calculated counting 3,000 cells, results are expressed as mean ± SD (n ≥ 3), ** p <0.01, * p <0.1; ANOVA and Dunnett's post-test. In the case of the trypomastigote release assay, HL-1 cells were infected and treated as described above. 72 hours later, medium was replaced with fresh prepared treatments until trypomastigotes were observed under microscope at six days post infection (pi). Supernatants were transferred to a new plate and quantification of trypomastigotes was performed with resazurin method. Results are expressed as mean ± SD (n ≥ 3). **** p<0.0001, t student test.

effector and a known target for PKA-mediated phosphorylation [26]. Since trypomastigotes released (Fig 3E), showed similar results than the percentage of infected cells and amastigotes/100cells (Fig 3C and 3D), we used this more reliable technique, in comparison to cell counting in the microscope, to evaluate the role of Rap1b phosphorylation. HL-1 cells overexpressing phospho-mimetic (S179D) or phospho-deficient (S179A) Rap1b mutants were infected and trypomastigotes released at day 6 pi were counted using resazurin method [27]. As shown in

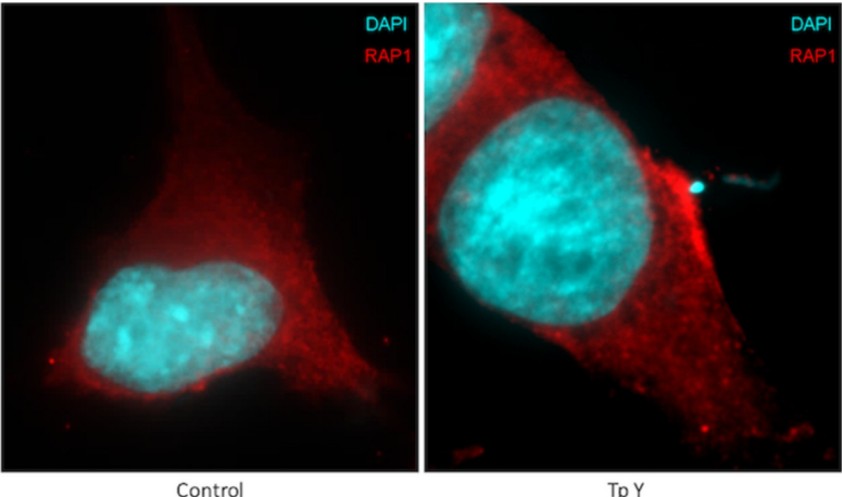

**Fig 4. Parasite-Rap1 colocalization.** For immunofluorescence, HL-1 cells were infected for 5 to 15 min with (Tp Y) trypomastigotes from *T. cruzi* Y strain (20:1 parasite to cell ratio) or mock infected (Control) and then fixed and incubated with primary antibody against Rap1 protein and a secondary antibody conjugated to Alexa594. Photos were taken with a fluorescence microscope.

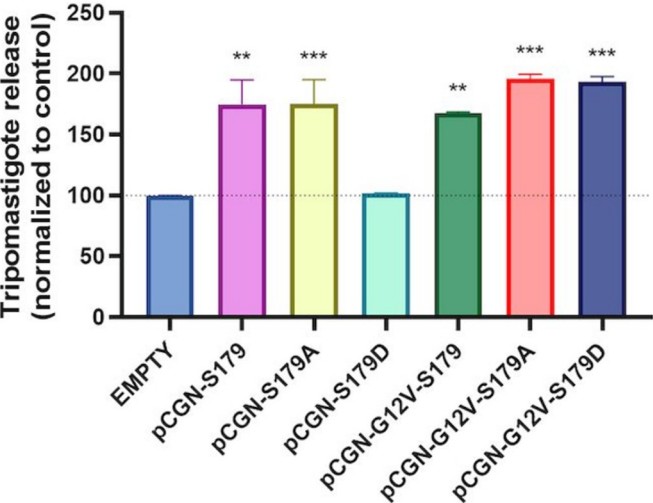

**Fig 5. PKA-dependent phosphorylation effect on parasite release.** Transfected HL-1 cells were infected and treated as described above. 72 hours later, medium was replaced with fresh prepared treatments until trypomastigotes were observed under microscope at six days post infection (pi). Supernatants were transferred to a new plate and quantification of trypomastigotes was performed with resazurin method. Infection of cells transfected with pCGN empty vector was considered as basal infection. Results are expressed as mean ± SD (n $\geq$ 3), *** p <0.001, ** p <0.01; ANOVA and Dunnett's post-test.

Fig 5, cells transfected with the phospho-mimetic Rap1b-S179D mutant presented a decrease in the number of released trypomastigotes, with respect to control cells or cells overexpressing the non-phosphorylable mutant Rap1b-S179A, supporting a PKA-dependent antagonistic effect on the pathway, as previously described in NRK cells [7]. Interestingly, the effect of phosphorylation could be reverted by transfecting cells with the double mutant G12V/S179D, a constitutive active phospho-mimetic Rap1, opening the possibility of a two-level regulation of PKA on the Epac/Rap1 pathway.

### MEK/ERK as a downstream effector of cAMP/Epac-mediated invasion of *T. cruzi*

In order to elucidate the involvement of MEK/ERK in the cAMP-dependent invasion, activation of ERK1/2 was analysed by Western Blot. An increase in ERK1/2 phosphorylation in both NRK (Fig 6A) and HL-1 (S5 Fig) cells was observed during the host cell infection. Interestingly, parasite-induced ERK phosphorylation was diminished by the addition of ESI-09 (S6 Fig). To determine whether the activation of ERK1/2 modulates the infection levels, cells pretreated with the MEK1/2 kinase inhibitor PD98059 were infected with the parasite. In accordance, the inhibition of ERK phosphorylation produced a significant decrease in infection (Fig 6B). MEK/ERK could be independently activated or a downstream effector of Epac/Rap1. The fact that the inhibition of MEK or Epac induced a similar decrease in invasion, and no additive or synergic effects were observed when both proteins were simultaneously inhibited (Fig 6C), suggests that MEK/ERK is a downstream effector of cAMP/Epac1/Rap1b-mediated invasion.

### Discussion

Infection of cell cultures shown to be a useful model to study host-cell interaction and invasion [28], however, it has to be considered that *T. cruzi* invasion showed to be a complex process just taking into account the different stages of the parasite with the ability to infect host cells.

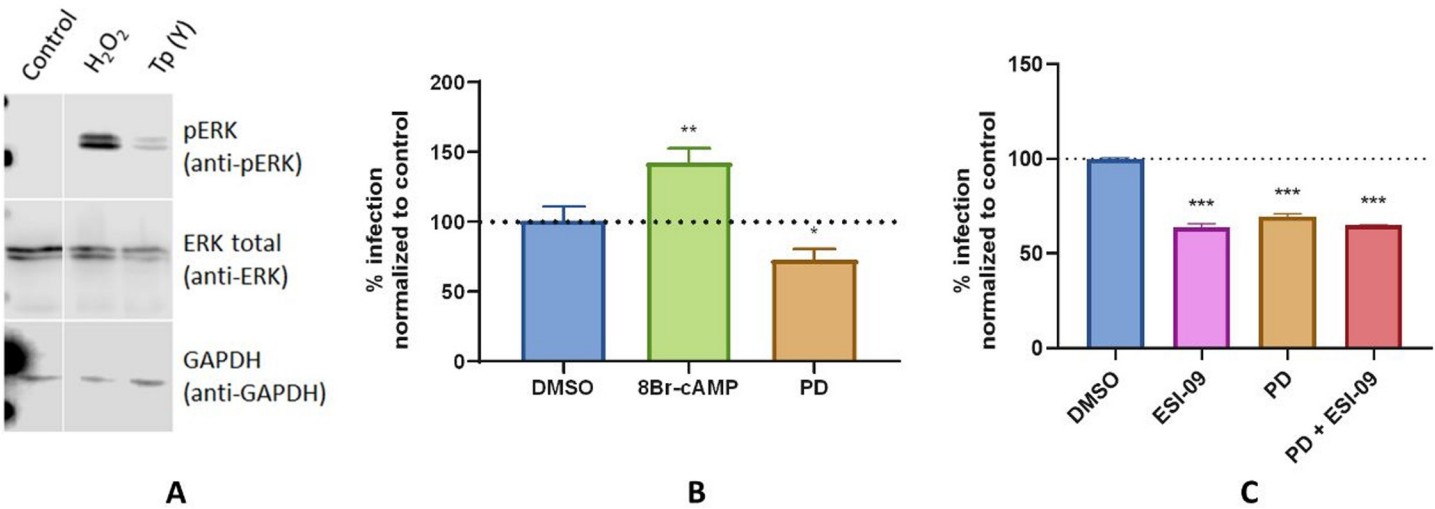

**Fig 6. ERK phosphorylation. A)** NRK cells were incubated for 2 h with trypomastigotes from *T. cruzi* Y strain (Tp Y), treated with 750 μM $H_2O_2$ for 5 min (positive control) or mock infected (Control). Then, cells were lysed and cracking buffer added for WB analysis. Representative WB performed on NRK cells is shown. **B)** and **C)** Invasion assay. Pretreated NRK cells were infected with trypomastigotes from *T. cruzi* Y strain (100:1 parasite to cell ratio) for 2 h, washed and incubated with fresh 10% FBS DMEM medium for 48 hs. Then, cells were fixed, stained with DAPI and percentage of infection determined by fluorescence microscopy. Infection of untreated cells was considered as basal infection. Results are expressed as mean ± SD ($n \geq 3$) *** $p < 0.001$, ** $p < 0.01$; ANOVA and Dunnett's post-test.

In addition, despite of being able to infect any nucleated cell *in vitro*, it has been shown *in vivo* that *T. cruzi* exhibits a certain cellular tropism [29] and that the signalling pathways activated in the host cell differ according to the stage of the parasite [30]. This complexity is even higher when considering different DTUs, strains and the repertoire of surface/secreted molecules that activates different signalling pathways in the host cell [1]. Additionally, differences in cell recognition and invasion, when performing assays in different cell lines, should be also consider [31]. With all these contemplations, and when it is clear that the possibility of replicating human biology in a dish is quite limited, *in vitro* models are an essential tool to dissect the phenomenon of parasite-host cell interaction and invasion.

In this context, it has been reported that the activation of cAMP-mediated signalling pathways triggers $Ca^{2+}$-dependent lysosomal exocytosis and promotes host cell invasion by *T. cruzi* [4]. The $Ca^{2+}$ release from intracellular compartments, such as the endoplasmic reticulum, is associated with an increase in intracellular levels of cAMP. In mammalian cells, cAMP downstream effectors, PKA and Epac, are involved in $Ca^{2+}$-activated exocytosis events [32]. Furthermore, members of these pathways, including Rap1, have been localized to late endosomes/lysosomes [15], and Epac-mediated activation of Rap1 has been identified in regulated exocytosis in human sperm [33], insulin secretion [34], and pancreatic amylase release [35]. It was previously shown that Epac1-mediated signalling represents the main mechanism for cAMP-mediated invasion by *T. cruzi* [7]. In addition, ERM proteins, which are essential for the function and architecture of the cell cortex by linking the plasma membrane to the underlying actin cytoskeleton [11], have been associated with the invasion of EAs [12]. Moreover, in confocal studies, it was shown that ERM proteins are recruited at the entry site of the parasites where they colocalize with F-actin, while its depletion inhibits HELA cells invasion [12]. Remarkably, one of its members, radixin, was identified as a scaffold unit for cAMP effectors in the spatial regulation of Epac1/Rap1-mediated signalling [9,10]. In this regard, we have previously revealed a link between Epac1 and radixin in the cAMP-mediated invasion of TCTs, by blocking the invasion of NRK cells with a permeable peptide of 15 amino acids that binds to the minimal ERM-binding domain of Epac [7]. In order to elucidate the role of cAMP

downstream effectors involved in *T. cruzi* invasion, we evaluated the activation of the cAMP/Epac pathway by TCTs of Y strain in NRK, HELA and HL-1 cell lines. NRK cells are normal fibroblasts from rat kidney, originally used in the establishment of cAMP as a modulator of invasion events [4] and to demonstrate the participation of Epac1 as the main effector of this modulation [7]. On the other hand, HELA cells are epithelial human cervix cells that have been widely used in invasion assays [36–38] and HL-1 cells, previously used in invasion assays, as well [39], are cardiomyocytes from mouse heart, one of the most important target organs in the infection and persistence of *T. cruzi*. Our data showed that the activation of the cAMP/Epac pathway by TCTs occurs regardless of the origin (rat, mouse, human) or the cell type (kidney, cervix, heart) that the parasite is invading. In addition, we investigated the role of Rap1b during the cAMP/Epac1-mediated invasion. Rap1b, a GTPase of the Ras family, is known to integrate Epac- and/or PKA-dependent events to achieve an efficient cAMP signal transduction [18,40,41]. Pull-down assays were used to detect higher levels of activated GTP-bound Rap1 in lysates from infected cells. Likewise, as shown in Fig 3, cells transfected with the constitutively active form of Rap1b (G12V) were more susceptible to infection, compared to the control. However, it is important to note that due to the fact that cells are constitutively overexpressing Rap1b (G12V), and cells were incubated for 48 hs after invasion in order to achieve sensitive in the parasite count, an effect of the overexpression of this small GTPase on parasite replication could not be excluded, and needs to be further explored. In addition to Rap1b activation, and in accordance to our hypothesis, data obtained from fluorescence microscopy assays evidenced the recruitment of Rap1b to the parasite entry site. Interestingly, when studying PKA participation using a specific inhibitors of this kinase, it was observed that the invasion levels of TCTs increased compared to the control [7], suggesting a PKA-dependent antagonist effect. This effect could be mediated by PKA phosphorylation of the effectors of the cAMP pathway, such as Epac and Rap1b. PKA-dependent phosphorylation at S179 of Rap1b has long been established [42]. Results presented in this work support the antagonistic effect of PKA through, at least, Rap1b phosphorylation, since trypomastigote release was affected in cells transfected with phospho-mimetic Rap1b-S179D, with respect to control cells and cells overexpressing Rap1b-S179A, the non-phosphorylable version of Rap1b. In line with these observations, it has been shown that Rap1b phosphorylation destabilizes the association of this protein with the plasma membrane and promotes Rap1b inactivation [43,44]. Nevertheless, as mentioned above, the experiments were carried out in transfected cells, and an effect of the overexpressed proteins on parasite replication could not be ruled out.

Overall, our data suggest that activation and relocalization of Rap1b are required as a mediator of the cAMP/Epac1 pathway during the TCT infection. In this scenario, the fact that PKA negative regulation on infection was abrogated in the presence of the constitutively active G12V mutation, suggests that Rap1b is required in the phosphorylated and inactive form to completely abolish the cAMP/Epac/Rap1b pathway of infection.

It has been described that the MEK/ERK pathway can be activated or inhibited by cAMP [45]. Furthermore, the activation of this pathway participates in the invasion of *T. cruzi* by way of the interaction of the host cell with parasite surface molecules, such as TS [46], Tc85 [47] or TSSAII [48]. Also, Rap1 is associated with the phosphorylation and activation of ERK1/2 in smooth muscle [19]. Accordingly, our data revealed that TCTs induce ERK1/2 phosphorylation in mammalian cells and ERK1/2 activation modulates the invasion of these parasites as a downstream effector of Epac/Rap1-mediated invasion.

Although the transient increase in cytosolic $Ca^{2+}$ concentration and lysosome recruitment that occur during invasion are characteristics shared between MTs and TCTs [4,5], the signalling pathways triggered by both forms of parasites in the host cell are different. In TCT invasion, ERK1/2 activation is a distinctive feature that is mediated by $Ca^{2+}$-dependent lysosomal

exocytosis through the regulation of F-actin and the activation of the focal adhesion kinase (FAK) [49]. During MT invasion, in contrast, PKC promotes $Ca^{2+}$ release from inositol 3-phosphate (IP3)-sensitive compartments through the binding of the surface glycoprotein gp82 to LAMP-2 receptors [30,50]. On the contrary, the activity of PKC is not required for the invasion of TCTs in NRK cells, since treatment with PKC inhibitors did not affect the response to $Ca^{2+}$ or the reorganization of F-actin, and has no effect on parasite internalization [51]. The divergence between the signalling pathways triggered by MTs and TCTs might be associated with the fact that the internalization of TCT is initiated by an invagination of the plasma membrane [52], in a lysosomal exocytosis-dependent process induced by a membrane injury and the following activation of the PMR mechanism [53]. These mechanisms lead to changes that take place through the inhibition of the Rho/Rho signaling pathway by PKA [54,55]. The fact that RhoA promotes actin polymerization but has a negative effect on EAs internalization during HELA cell invasion [56] and that Rap1b inhibits RhoA/ROCK activity in the muscle smooth tissue [57], suggest the hypothesis that the cAMP/Epac1/Rap1b signalling pathway could be activated in the first steps of the invasion by *T. cruzi*, promoting $Ca^{2+}$-dependent lysosomal exocytosis and the reorganization of the cytoskeleton. Once the parasite is inside the cell, a PKA-mediated inhibition of Epac/Rap1b might be necessary for the parasite retention. In accordance, our results showed that Rap1b seems to be associated with the plasma membrane at the parasite entry site where it could be required during the internalization process and PKA had an antagonistic effect, probably through the phosphorylation of the S179 of Rap1b.

In this work, we have gathered evidence strongly suggesting that the cAMP/Epac/Rap1b/ERK pathway is activated during the early steps of host cell infection and that it would be negatively regulated by PKA, possibly through the phosphorylation of Epac and/or Rap1b. Importantly, a detailed characterization of effectors involved in *T. cruzi* invasion would provide an attractive set of new therapeutic targets for the repositioning or the development of new antiparasitic drugs, since there is a large variety of therapies that target cAMP-mediated signalling [58].

## Supporting information

**S1 Fig. Densitometry analysis.** Chemiluminescence was recorded with the C-DiGit scanner (LI-COR), and bands were quantified and normalized against the input using ImageJ and ImageLab 6.1 (Bio-Rad) software. The normalization was performed following the "Western Blot Normalization Using Image Lab Software" guide. Results are expressed as mean ± SD (n≥3). * p<0.05, ** p <0.005, One-way ANOVA–Dunnett´s multiple comparison test.
(PDF)

**S2 Fig. Rap1b pull-down assays. A)** HA-Rap1 transfected HELA cells were incubated for 2 h 37.5uM ESI-09 or 0.1% DMSO. Then, cells were lysed and pull-down assay with glutathione-agarose resin performed for 1 h at 4˚C. Resin was washed and eluted with cracking buffer for WB analysis. **B)** Bands were quantified and normalized against the input using ImageJ cell software. Results are expressed as mean ± SD (n≥3). ** p<0.01, t student test.
(TIFF)

**S3 Fig. Immunofluorescence of HL-1 cells.** A) Cells were infected for 5 to 15 min with (Tp Y) trypomastigotes from *T. cruzi* Y strain or mock infected (Control) and then fixed and incubated with primary antibody against Rap1 protein and a secondary antibody conjugated to Alexa594. B) Line profiles obtained from the line shown in the Rap1 quadrant. Photos were taken with a fluorescence microscope. Only a representative image is shown. Scale bar: 10 μm.
(PDF)

**S4 Fig.** A) HeLa cells were incubated with trypomastigotes and the percentage of infected cells with Rap1 positive signal at the site of parasite entry were quantified at 5 and 20 minutes. Quantification was performed counting at least 30 infected cells of each time point in 3 independent experiments. $^*$ p < 0.01 (t test). B) Immunofluorescence of HL-1 cells were infected for 20 min with trypomastigotes from *T. cruzi* Y strain (Tp Y) (20:1 parasite to cell ratio) or mock infected (Control) and then fixed and incubated with primary antibody against Rap1 protein and a secondary antibody conjugated to Alexa594. Photos were taken with a fluorescence microscope.
(TIF)

**S5 Fig. Raw data ERK phosphorylation.** Upper) NRK cells were incubated for 2 h with trypomastigotes from *T. cruzi* Y strain (Tp Y), treated with 750 μM H2O2 for 5 min (positive control) or mock infected (Control). Then, cells were lysed and cracking buffer added for WB analysis. Lower) HL-1 cells were incubated for 2 h with trypomastigotes from *T. cruzi* Y strain (Tp Y) or mock infected (Control). Then, cells were lysed and cracking buffer added for WB analysis.
(PDF)

**S6 Fig.** A) HeLa cells were pretreated with H2O2 (750 mM, 5 min) or ESI-09 (37.5mM, 30 min), then cells were washed and incubated with trypomastigotes for 2 hours. Cells were washed and lyzed. WB with ERK, P-ERK or GAPDH antibodies were performed. B) Erk and P-Erk band density were normalized to GAPDH, then P-Erk expression was relativized to total Erk. Band density were quantified using ImageJ. $^*$ p < 0.01 (t test).
(TIFF)

## Author Contributions

**Conceptualization:** Martin M. Edreira.

**Formal analysis:** Gabriel Ferri, Daniel Musikant, Martin M. Edreira.

**Funding acquisition:** Martin M. Edreira.

**Investigation:** Daniel Musikant, Martin M. Edreira.

**Methodology:** Gabriel Ferri, Martin M. Edreira.

**Project administration:** Martin M. Edreira.

**Supervision:** Martin M. Edreira.

**Validation:** Gabriel Ferri.

**Writing – original draft:** Gabriel Ferri, Martin M. Edreira.

**Writing – review & editing:** Martin M. Edreira.

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
