## [Decision Letter · Decision Letter 0]

9 Aug 2022

Dear Dr. Edreira,

Thank you very much for submitting your manuscript "Host Cell Rap1b mediates cAMP-dependent invasion by Trypanosoma cruzi" for consideration at PLOS Neglected Tropical Diseases. As with all papers reviewed by the journal, your manuscript was reviewed by members of the editorial board and by several independent reviewers. In light of the reviews (below this email), we would like to invite the resubmission of a significantly-revised version that takes into account the reviewers' comments. 

We cannot make any decision about publication until we have seen the revised manuscript and your response to the reviewers' comments. Your revised manuscript is also likely to be sent to reviewers for further evaluation.

Sincerely,

Luisa Magalhães

Academic Editor

Ricardo Fujiwara

Section Editor

Reviewer's Responses to Questions

**Key Review Criteria Required for Acceptance?**

**Methods**

-Are the objectives of the study clearly articulated with a clear testable hypothesis stated?

-Is the study design appropriate to address the stated objectives?

-Is the population clearly described and appropriate for the hypothesis being tested?

-Is the sample size sufficient to ensure adequate power to address the hypothesis being tested?

-Were correct statistical analysis used to support conclusions?

-Are there concerns about ethical or regulatory requirements being met?

Reviewer #1: The methods are satisfactory and include commercial information on reagents used. Nonetheless, a few points need to be reviewed, for example: 

1) Although the figures’ subtitles describe the sample size, it is important to describe them in the methods as well. It is important to express if the experiments are independent and if each experiment was done in triplicate.

2) Some of the information needs to be reported in the methods, for instance, the g force, and how much time was used in centrifugation for obtaining the trypomastigotes (line 84). In line 86, it is important to describe the size of the glass cover slides. 

3) In section “GST Pull-down” (line 111), the relationship between glutathione and the question is not clear. It is crucial to explain how this methodology works and the outcome’s relation to the question.

4) The Resazurin sodium salt dye is commonly used as a fluorogenic oxidation-reduction indicator in a variety of cells. Therefore, it is necessary to describe how this reagent was used in your experiments to quantify the trypomastigotes released in the supernatant of the experiments.

Reviewer #2: (No Response)

Reviewer #3: The main goal of the manuscript is to describe the pathway by which cAMP/Epac participate in T.cruzi TCTs mechanism of host cell entry. Based on previous results from the group and the literature, the main hypothesis is that the pathway involves the activation of Rap1b, an effector downstream of Epac pathway. So the objectives are clear and most of the experimental design is suited to test this main hypothesis. There is, though, one experiment that was not suited to show the point that authors wanted to test. They have used a step at the end of parasite intracellular life cycle (trypomastigote release from infected cells) to indirectly evaluate the role of rap1b in parasite invasion, which is not suited to show this point, since there are direct methods, used in the present manuscript , which would incontestably prove this point. Some of the statistical analyses (figure2, for example) could be reviewed, since there are differences that should be statistically significant and do not show significancy with the test used. All the other points, such as sample size and methods description are ok.

**Results**

-Does the analysis presented match the analysis plan?

-Are the results clearly and completely presented?

-Are the figures (Tables, Images) of sufficient quality for clarity?

Reviewer #1: The results were well-presented and were described adequately. The figures’ subtitles were also clear, but in the caption for figure 4, the ratio of the parasite to cell was missing. Moreover, the NRK graphs in figure 1 require 8-Br-AMPc to be changed to 8-Br-cAMP.

Reviewer #2: (No Response)

Reviewer #3: All the experiments were designed to show the points they want to prove and, as mentioned, are adequate to make their point, except for figure 5. The data is well presented, even though there are some important points of discussion, which will be addressed in the general comments. The figures are well presented and of sufficient quality for clarity.

**Conclusions**

-Are the conclusions supported by the data presented?

-Are the limitations of analysis clearly described?

-Do the authors discuss how these data can be helpful to advance our understanding of the topic under study?

-Is public health relevance addressed?

Reviewer #1: The conclusions are favorable and are supported by the data presented in the article. Although the experimental design using cell lines is adequate, the manuscript would benefit from discussing the limitations of the use of cell lineage to answer the questions presented.

Reviewer #2: (No Response)

Reviewer #3: Part of the conclusions are supported by the data presented. There are some relevant points, though, that need to be addressed before a final conclusion is made. The discussion of the data is relevant and address the important points of the study for the field.

**Editorial and Data Presentation Modifications?**

Reviewer #1: The article is useful to understand more about the mechanism of T. cruzi invasion, a fundamental aspect of Chagas disease, which is still not completely clear. Moreover, a vaccine for this disease has still not been developed, and the treatment is extremely toxic for the patients. Thus, to know more about the possibilities of intervention in the relationship between T. cruzi and cells becomes an interesting point of investigation. 

The main point that needs to be reviewed is the introduction. 

The introduction of the article “Host Cell Rap1b mediates cAMP-dependent invasion by Trypanosoma cruzi” needs to be rewritten. The objectives and hypothesis are not clear. Furthermore, it is necessary to put in the introduction more information about the cAMP/Epac/Rap1 pathway, especially about Rap1b, which is one of the study’s questions. The information presented in lines 194-217 is important to understand the article, and it would be enlightening if presented in the introduction with this clarity and organization.

Reviewer #2: (No Response)

Reviewer #3: no comments

**Summary and General Comments**

Reviewer #1: The study entitled “Host Cell Rap1b mediates cAMP-dependent invasion by Trypanosoma cruzi” is essentially about the importance and participation of some molecules in Trypanosoma cruzi host cell invasion. Using immunofluorescence and western blot, the authors demonstrate the involvement of the MEK/ERK signaling downstream of cAMP/Epac/Rap1b-mediated invasion in different cell line hosts. Understanding this invasion mechanism and possible interventions is of great value for Chagas disease. With some adjustments, the introduction will be more directed to the study proposed by the authors, and even more understandable for the reader, as already mentioned. The information on some points of the methodology is important so that other researchers can reproduce the experiments and so that readers can understand the techniques used. The results obtained corroborate the study's hypothesis and questions (even if they are not completely clear), and the conclusions allow the reader to understand the importance of this study, although it was conducted in cell lines and this fact has not been discussed.

Reviewer #2: Ferri and Edreira sought to evaluate the participation of Rap1b and its GEF, Epac (cAMP-activated), on host cell invasion by tissue-cultured derived trypomastigotes of Trypanosoma cruzi.

 Authors provide however insufficient results to fully support this hypothesis.

The manuscript demands substantial improvement on quality and on experimental evidences to be considered for publication and to make an impact in the community.

Major concerns below:

Line 145: “high levels of cAMP induced by..” no experiment showing increased levels of cAMP was presented. Authors could show the dose-dependent response in host cell cAMP levels upon treatment with these drugs and correlate it with parasite internalization.

Line 149: Participation of Epac on T. cruzi invasion should be further corroborated by another approach such as Epac protein knockdown or knockout techniques.

Line 155: Epac inhibitor, ESI-09, should also be included in these studies to address eventual reduced activation of Rap1b.

Line 157: Different experiments can be explored in different paragraphs/topics.

Line 160: it was not clear why and how the trypomastigote release experiment was done. Were authors aiming to address Rap1b also in parasite intracellular life cycle (multiplication and egress)?

Line 161: It is not clear what the G12V stands for.

Line 162: “…played a role in the early steps of the establishment of infection…” Can the authors elaborate on that? Do authors suggest that Rap1b, besides invasion, also modulate host Rap1b pathway once parasites are internalized?

Line 165: Quantification of Rap1b recruitment to the parasite’s invasion sites is needed. Also to be included brightfield channels to spatially correlate parasite-host cell contact zones with Rap1b recruitment. Also, how are these Rap1b recruitments correlated to lysosomes and actin, major drivers of parasite internalization?

Line 167: Authors comment that Rap1b needs to be activated to be recruited by the parasites. To further corroborate this hypothesis authors should include activators and inhibitors of the pathway as in Fig1 (8-Br-cAMP and ESI-09) and do proper quantifications of Rap1 recruitment.

Line 174: “To evaluate the role of Rap1b phosphorylation…” in parasite infection? If so, assays as in Fig1 should be performed instead of the trypomastigote release showed.

Line 186: ERK activation should be assessed in time-dependent manner upon T. cruzi-host cell interaction because the activation profile can differ between infective (trypomastigote and amastigote) and non-infective (epimastigote) forms (Bonfim-Melo et al., 2015, Ferreira et al., 2016).

Line 193: I agree no additive effects on double inhibition of Epac and ERK further suggests they act on the same pathway. However further experiments could be performed to further corroborate that Epac acts upstream ERK during parasite internalization. For example: does ESI-09 inhibit the parasite-induced ERK activation?

Fig1 (and all others): It is recommended to use data presentation with individual observations rather than bar graphs. It helps readers to understand for example why it was observed statistical differences in cases of small differences in group means (like in Fig1 NRK) while no statistical difference in cases of big differences in group means (like in Fig2 NRK and HeLa 8Br-cAMP).

Fig2: Most WB images are of very poor quality (Fig2 “HL-1”) and with cropped bands. It is highly recommended to present uncropped versions of these data.

Fig2A: The input panel from HeLa cells shows no consistent Rap1b between the groups. How many experiments are these representative images from? Authors should also provide densitometry analyses of these bands as well as the original images for the cropped and uncropped blots.

Figure3: It is not clear what i, ii and iii stand for. I recommend using A to E labels for all plots.

Fig4: It should be shown a couple of parasite-host cell interactions with an inset detailing the most representative one. Also, a scale bar should be included.

Fig5: figure title states “…effect on invasion.” But apparently the experiment quantifies supernatant release amounts of trypomastigotes after 6 days of infection; this is not directly measuring parasite internalization rather it assesses construct effects on parasite intracellular life cycle/multiplication.

Fig5: It is not clear which group was used as control for the statistics.

Fig6: (A) Very poor-quality blots, there are bands from neighbouring lanes disturbing proper interpretation of the desired bands. It is also not clear which cell line was specifically used for in this blot.

Fig6B: Legend should clearly state the parasites were incubated for 2h with host cells but not 48 h.

Methods: Assessment of invasion fitness should be performed after parasite removal (after 2h of parasite incubation). Further incubation of the cells for 48 h allows parasites to start intracellular replication leading to cofounding interpretations. Authors should elaborate why they have included this unusual step of 48 h incubation.

Reviewer #3: The main goal of the manuscript is to describe the pathway by which cAMP/Epac participate in T.cruzi TCTs mechanism of host cell entry. Based on previous results from the group and the literature, the main hypothesis is that the pathway involves the activation of Rap1b, an effector downstream of Epac pathway.

The authors do show the ubiquitous activation of cAMp for different cell types. However, when they address the role of Rap1b in the cAMP/Epac pathway there are some important points in the data presented that need to be addressed and they are listed below:

1. The authors say in the text that “As shown in Figure 2, higher levels of activated GTP-bound Rap1 were detected in lysates from cells incubated with 8-Br-cAMP and trypomastigotes, supporting the involvement of Rap1b in cAMP-mediated invasion.” However statistically significant results are only observed for Hela cells exposed to T. cruzi Y strain or HL-1 treated with BrcAMP or exposed to Tc Y strain. The results for NRK cells are not statistically different from control cells. It is a fact that the bars do seem to show some differences, but the statistics did not say the same. So at least this sentence need to be rephrased. Later to prove the importance of Rap1b activation for T.cruzi host cell entry they used cells expressing a constitutively active form of Rap1b. However, they only use Hela and HL-1 for these experiments and no tests with NRKs are shown. If they really want to make a point that this pathway is ubiquitous for all cell lines, they need to perform these experiments also with NRKs. For these experiments they evaluate not only parasite cell invasion, but also the number of intracellular amastigotes and number of released parasites. This last one (# released parasites) though was only performed for HL-1 cells. Why? The same result was obtained for Hela? These data should be presented and discussed in the manuscript. In addition, although the number of released parasites may reflect the amount of cell infection/infected cells, there are a number of manuscripts in the field showing that invasion and intracellular multiplication are two different events and that released parasites are more related to the latter other than necessary to cell invasion. So authors need to be very careful to assume that number of released parasites are in fact a good measure of invasion levels.

2. In figure 4 the authors show a fluorescence microscopy assay, showing the co-localization of the parasite with Rap1b at the site of entry. How frequent this was observed? A quantification needs to be included in the figure. The assays were performed with HL-1 cells. Was this also observed for NRKs and Hela? Again in order to make an statement that this is an ubiquitous process for all types, this data should be included.

3. As mentioned before, invasion and intracellular multiplication are not necessarily related and there are data in the literature showing that parasite intracellular multiplication can be modulated by ROS, for example, so using parasite release is not a good method to test invasion. Thus, the results presented in figure 5, where authors test the influence of activated and not activated Rap1b in T. cruzi infection, should be performed using a classical invasion assay and not the number of released parasites. In this sense, the sentence “As shown in Figure 5, cells transfected with the phospho-mimetic Rap1b-S179D mutant presented a decreased invasion with respect to control cells or cells overexpressing the non-phosphorylable mutant Rap1b-S179A, supporting a PKA-dependent antagonistic effect on the pathway” is not what the results show. This correlation cannot be done and, as mentioned, in order to make this affirmative they need to perform an invasion assay showing the same results. It is possible that Rap1b may also interfere with parasite intracellular multiplication.

PLOS authors have the option to publish the peer review history of their article (what does this mean?). If published, this will include your full peer review and any attached files.

Reviewer #1: Yes: Carolina Cattoni Koh

Reviewer #2: No

Reviewer #3: No
---

## [Decision Letter · Decision Letter 1]

27 Dec 2022

Dear Dr. Edreira,

Thank you very much for submitting your manuscript "Host Cell Rap1b mediates cAMP-dependent invasion by Trypanosoma cruzi" for consideration at PLOS Neglected Tropical Diseases. As with all papers reviewed by the journal, your manuscript was reviewed by members of the editorial board and by several independent reviewers. In light of the reviews (below this email), we would like to invite the resubmission of a significantly-revised version that takes into account the reviewers' comments. 

We cannot make any decision about publication until we have seen the revised manuscript and your response to the reviewers' comments. Your revised manuscript is also likely to be sent to reviewers for further evaluation.

Sincerely,

Luisa Magalhães

Academic Editor

Ricardo Fujiwara

Section Editor

Reviewer's Responses to Questions

**Key Review Criteria Required for Acceptance?**

**Methods**

-Are the objectives of the study clearly articulated with a clear testable hypothesis stated?

-Is the study design appropriate to address the stated objectives?

-Is the population clearly described and appropriate for the hypothesis being tested?

-Is the sample size sufficient to ensure adequate power to address the hypothesis being tested?

-Were correct statistical analysis used to support conclusions?

-Are there concerns about ethical or regulatory requirements being met?

Reviewer #1: Describe the concentration of DMSO used (line 179).

Reviewer #3: no new comments

**Results**

-Does the analysis presented match the analysis plan?

-Are the results clearly and completely presented?

-Are the figures (Tables, Images) of sufficient quality for clarity?

Reviewer #1: no comments

Reviewer #3: There are some points that still need to be addressed and will be listed in the general comments

**Conclusions**

-Are the conclusions supported by the data presented?

-Are the limitations of analysis clearly described?

-Do the authors discuss how these data can be helpful to advance our understanding of the topic under study?

-Is public health relevance addressed?

Reviewer #1: no comments

Reviewer #3: There are some points that still need to be addressed and will be listed in the general comments

**Editorial and Data Presentation Modifications?**

Reviewer #1: On the Y axes of the figures it is written: normalized or relative to something . For me, the analysis and meaning of both is the same. I suggest that it be standardized.

Reviewer #3: (No Response)

**Summary and General Comments**

Reviewer #1: no comments

Reviewer #3: The authors have significantly improved the manuscript, addressing the points raised in the first review. However, there are still some relevant points to be addressed.

-In figure 3, again, after 48h you already have parasite multiplication. To attest that this is related to invasion, they need to analyze the cells right after parasite exposure. Also, it is not clear what they are calling invasion. Is it the number of infected cells? This should be clearer in the methods and in the legend of the figure. It would be important, since in the way they presented the results (relative to non-transfected) we cannot have an idea of the number of parasites/cell, which would indicate whether RAP1b does interfere with parasite multiplication. This clarification and discussion is important, since the results could still be related to parasite multiplication, rather than only invasion. Multiplication alone could account for more parasites being released from the culture. 

- Still concerning Figure 3, as a response to a comment from the first review the authors mentioned they used only HL-1 for parasite release, because it is a cardiomyoblast, a cell type important for Chagas disease. Also, because this had already been shown previously for NRKs.

It should be also mentioned in the text that there is already published data showing similar results for NRKs and also include a supplementary figure showing the preliminary data for Hela. This is important to prove the point that this pathway is in fact ubiquitous. With the results shown, they can only speculate that, due to some data in the literature, as well as preliminary results, this may be an ubiquitous pathway.

-Concerning figure 4, results with Hela should be mentioned in the text and included as a new supplementary figure. This is important to strengthen the point that this is a ubiquitous process. Otherwise, the authors should revise this statement and say that they observed it only for HL-1 and that, according to preliminary data, it might work the same way for other cell types. Supplementary figure 4 should be included as a new panel in this figure.

-In response to the comment made to figure 5 and the affirmative that due to more released parasites, the invasion rates were higher. The authors replied that this statement was valid, since “previously reported results with PKA inhibitors and activators showed similar results”. This is not the case. In order to make that affirmative “there was more invasion” it is needed to be shown, that there are more internalized parasites/ infected cells or even more cells infected at time 0, right after parasite exposure. They cannot use “more invasion” if what they are showing is “more parasite release”. The fact that there are data in the field that could corroborate this idea, does not allow them to assume this is true, they could only speculate that this might be the case and include the mentioned reference. This must be changed in the text and the legend of the figure, as well as in the discussion of the data.

PLOS authors have the option to publish the peer review history of their article (what does this mean?). If published, this will include your full peer review and any attached files.

Reviewer #1: No

Reviewer #3: No
---

## [Decision Letter · Decision Letter 2]

22 Feb 2023

Dear Dr. Edreira,

We are pleased to inform you that your manuscript 'Host Cell Rap1b mediates cAMP-dependent invasion by Trypanosoma cruzi' has been provisionally accepted for publication in PLOS Neglected Tropical Diseases.

Best regards,

Luisa Magalhães

Academic Editor

Ricardo Fujiwara

Section Editor

Reviewer's Responses to Questions

**Key Review Criteria Required for Acceptance?**

**Methods**

-Are the objectives of the study clearly articulated with a clear testable hypothesis stated?

-Is the study design appropriate to address the stated objectives?

-Is the population clearly described and appropriate for the hypothesis being tested?

-Is the sample size sufficient to ensure adequate power to address the hypothesis being tested?

-Were correct statistical analysis used to support conclusions?

-Are there concerns about ethical or regulatory requirements being met?

Reviewer #3: no new comments

**Results**

-Does the analysis presented match the analysis plan?

-Are the results clearly and completely presented?

-Are the figures (Tables, Images) of sufficient quality for clarity?

Reviewer #3: All the points raised in the second review have been appropriately answered

**Conclusions**

-Are the conclusions supported by the data presented?

-Are the limitations of analysis clearly described?

-Do the authors discuss how these data can be helpful to advance our understanding of the topic under study?

-Is public health relevance addressed?

Reviewer #3: All the points raised in the second review have been appropriately answered

**Editorial and Data Presentation Modifications?**

Reviewer #3: no comments

**Summary and General Comments**

Reviewer #3: All the points raised in the second review have been appropriately answered.

PLOS authors have the option to publish the peer review history of their article (what does this mean?). If published, this will include your full peer review and any attached files.

Reviewer #3: No

---

## [Editor Report · Acceptance letter]

6 Mar 2023

Dear Dr. Edreira,

We are delighted to inform you that your manuscript, "Host Cell Rap1b mediates cAMP-dependent invasion by Trypanosoma cruzi," has been formally accepted for publication in PLOS Neglected Tropical Diseases.

Best regards,

Shaden Kamhawi

co-Editor-in-Chief

Paul Brindley

co-Editor-in-Chief
